# Quantifying Water Provision Service Supply, Demand, and Spatial Flow in the Yellow River Basin

**Yang Liu [1], Yang Yang [2], Zhijie Wang [3] and Shaoshan An [1],***

1 State Key Laboratory of Soil Erosion and Dryland Farming on the Loess Plateau, Northwest A&F University, Yangling, Xianyang 712100, China
2 State Key Laboratory of Loess and Quaternary Geology, Institute of Earth Environment, Chinese Academy of Sciences, Xi'an 710061, China
3 College of Life Sciences, Guizhou University, Guiyang 550025, China
* Correspondence: shan@ms.iswc.ac.cn

**Abstract:** Quantifying and spatial mapping the ecosystem services driven by land use change will help better manage land and formulate relevant ecological protection policies. However, most studies to date just focused on water supply services, and ignore water demand services and their supply–demand coupling mechanisms. Ecosystem service flow could be used to evaluate the imbalance between water supply and demand. Therefore, this study takes the Yellow River Basin as the research object to quantify the supply, demand, and spatial flow of water provision services. The results showed that land use and land cover (LULC) played a critical role in the spatial distributions of water supply and demand in the Yellow River Basin. The total water supply was $3.03 \times 10^{11}$ m$^3$, with a range of $3.29 \times 10^8$ m$^3$ to $7.35 \times 10^{10}$ m$^3$ for different sub-watersheds. The spatial patterns of water supply were strongly different from those in water demand, resulting in obvious spatial mismatches. There was a higher water demand for constructional areas and agricultural lands, which had relatively lower water supply. Most water areas and natural lands provide much more water supply than demand. We used a water flow process to assess the water provision service between water supply side and demand side. The water flow process suggested that the Yellow River Basin had an obvious imbalance between water supply and demand depending on land use and populations, which would help policy makers to manage water resources through optimizing land management in different cities and finally achieving a balance between water supply side and demand site.

**Keywords:** ecosystem services flow; water supply; water demand; InVEST model; the Yellow River Basin

## 1. Introduction

Land use change is cumulatively considered the most important driving force in global environmental change [1]. A central challenge for sustainability is how to preserve water resources and the services that they provide us while enhancing food production to meet the needs of the increasing population. With the rapid development of human society, the imbalance between food supply and demand caused by population growth has led to major changes in land use and land cover (LULC), especially for the expansion of agricultural land and urbanization [1–3]. At the same time, urban expansion and economic development have caused major changes in LULC [4,5]. Changes in land use lead to uneven distribution of water resources, cause imbalances between water supply and demand, and other problems, seriously restricting the sustainable development of mankind [6]. Water shortage most severely restricts socio-economic development [7], especially in urban areas and arid and semi-arid areas (i.e., northwest China).

Land management is a response to a series of environmental problems caused by climate change [8] and anthropogenic activities, such as the imbalance of water supply and

demand. By evaluating the ecosystem services driven by land use change, it is possible to effectively optimize the allocation of limited land resources, improve the ecological environment, and achieve the harmonious development of economy, society, and ecology. Land use changes affect the imbalance between water supply and demand [9]. For example, the expansion and high intensity use of agricultural land can lead to an increase in irrigation water, especially in the dry season and arid areas; industrialization and urbanization can also aggravate the increase in urban water use to achieve high income. Therefore, changes in land use lead to serious conflicts between the supply and demand of water resources in some areas. Reasonable land use could make full use of water resources and maintain the balance between supply and demand. In addition, land use changes could be used to effectively explore the driving mechanism of changes in ecosystem services.

To better characterize the relationship between supply and demand of ecosystem services, we introduce ecosystem service flows to quantify the spatial flow of ecosystem services. Ecosystem service flow characterization is the spatiotemporal relationship between ecosystem service in supply and demand. So far, most of the current studies are based on the characteristics of temporal and spatial changes in the supply of ecosystem services, ignoring the demand for ecosystem services. Compared with other ecosystem service flows, relevant research has been gradually carried out on ecosystem water supply service flows.

A previous study assessed the water supply service flow along the river basin [10], and the results showed that land use change is the main driving factor resulting in the imbalance of supply and demand. Li et al. (2017) used the InVEST model to link historical water consumption statistic data to evaluate the balance between supply and demand of freshwater resources from 2000 to 2010 in China [11]. However, most researchers ignored the associations between water supply and demand and include the mismatch in the scale of supply and demand [12]. In addition, few studies determined the relationships between water supply and demand considering the impact of upstream water. Therefore, Chen et al. (2020) proposed a framework to illustrate the flow of water supply services in the Yanhe watershed [10], which is included in the Yellow River Basin. In order to quantify the ecosystem service flow of the whole river, we evaluated water supply, demand, and spatial flows of water supply services based on land use changes in the Yellow Basin according to the framework developed by previous study [10], which would fill this research gap about the decoupling relationships between water supply and demand. We aim to (1) assess the spatial patterns of water supply and demand of the Yellow River Basin; (2) evaluate the flow process between water supply and demand; (3) analyze the main drivers of changes between water supply and demand, and provide some useful suggestions for the government to manage the limited water resources in Yellow River Basin.

## 2. Materials and Methods

### 2.1. Studying Area

This study was conducted within the Yellow River Basin (31°31′ N~43°31′ N, 89°19′ E~119°39′ E), which is the fifth largest river in the world and the second largest river in China. It occupies 19% (79,500,000 ha) of China's land area and significantly affects local ecological functions, ecosystem services, and human well-being. It spans eight provinces (Figure 1), including Inner Mongolia, Shanxi, Qinghai, Gansu, Ningxia, Henan, and Shandong. It contributes 14.7% of the whole population and 14.5% of GDP (Gross Domestic Product) in China. It plays such critical important roles for mankind that the Chinese government proposed a national strategy of ecological protection and high-quality development in the Yellow River Basin in 2019. The Yellow River Basin is the most important water pool in the North China, providing all kinds of ecosystem services, including provisioning, regulating, cultural and supporting services. Therefore, the Yellow River Basin has become a key zone for the development of ecological protection and sustainability in China.

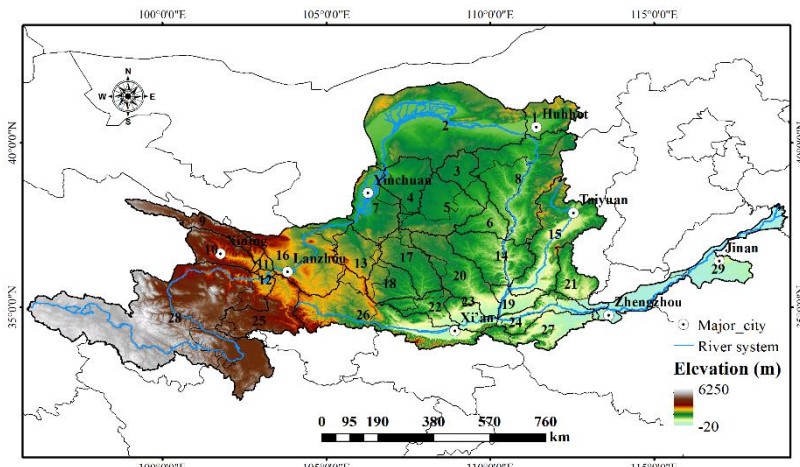

**Figure 1.** The location and the sub-watersheds of the Yellow River Basin. The number is the sub-watershed in the whole river.

The Yellow River Basin spans 1900 km from east to west and 1100 km from north to south. The annual average precipitation varies from 116 to 1038 mm, and the annual temperature changes from −13 °C to 15 °C with an obvious spatial pattern. The southeast areas show relatively higher temperature and precipitation, while the northeast areas show relatively lower temperature and precipitation.

### 2.2. Data Collection

The datasets include a Digital Elevation Model (DEM), meteorological, land use and land cover (LULC), and soil and water demand data, and the data resources are listed in Table 1. The meteorological data used mainly include rainfall, temperature, and other data from 752 meteorological stations, which were interpolated in ArcGIS 10.2 to obtain the meteorological datasets of the Yellow River Basin (Figure 2). The meteorological dataset was downloaded from the Meteorological data Service Centre (http://data.cma.cn/, accessed on 10 May 2022), and the time was from January 1992 to December 2020. The final meteorological data was the annual means to improve the accuracy. The LULC data from 2010 was collected from the Resource and Environmental Science Data Center of the Chinese Academy of Sciences (http://www.resdc.cn, accessed on 10 May 2022). In this study, the LULC had a spatial resolution of 300 m × 300 m. The original LULC datasets were reclassed into 6 categories, including farmland, forest, grassland, water area, construction area, and bare land. The DEM data was collected from Geospatial Data Cloud (http://www.gscloud.cn/, accessed on 10 May 2022). The original resources of basic data were listed in Table 1. The sub-watersheds were generated from the Hydrology Analyst Tools in ArcGIS 10.2. The Yellow River Basin was divided into 29 main sub-watersheds (Figure 2).

**Table 1.** The sources of the required data.

| Required Data | Description | Source |
|---|---|---|
| Topographical data | Digital Elevation Model (DEM) | Geospatial Data Cloud (http://www.gscloud.cn/, accessed on 10 May 2022) |
| Land use and Land cover (LULC) | Agriculture, Forest, Grassland, Water, Urban and Unused land (2010) | Data Center for Resources and Environmental Sciences, Chinese Academy of Sciences (http://www.resdc.cn/, accessed on 10 May 2022) |

**Table 1.** *Cont.*

| Required Data | Description | Source |
|---|---|---|
| Soil properties | Soil clay, sand, silt, and soil organic matter | Harmonized World Soil Database version 1.1 (HWSD) (http://westdc.westgis.ac.cn/, accessed on 10 May 2022) |
| Meteorological data | Temperature and Precipitation (1992–2020) | Meteorological data Service Centre (http://data.cma.cn/, accessed on 10 May 2022) |
| Statistical data | Water use (Agricultural water, Industrial water, Domestic water, and Forest-herd-fishing water), Population, GDP, etc. | Water Authority and Bureau of Statistics |

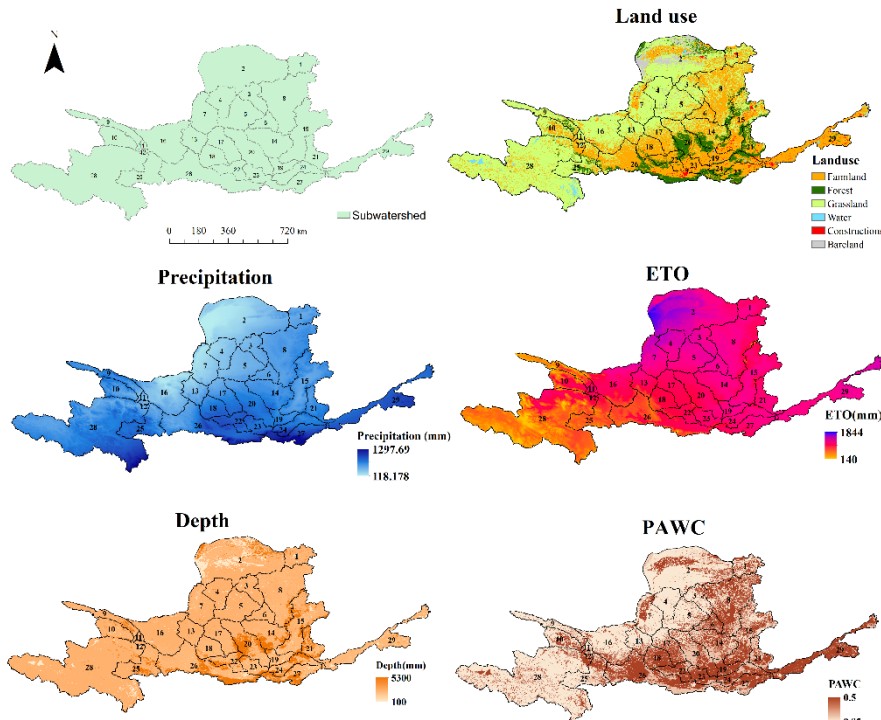

**Figure 2.** The basic characteristics (sub-watersheds, land use and land cover (LULC), annual precipitation, reference evapotranspiration (ETO), root depth, and the available volumetric water content for the plant (PAWC) of water yield) for the Yellow River Basin. (The numbers from 1 to 29 mean serial number of sub-watersheds).

*2.3. Data Analysis*

The InVEST model (https://naturalcapitalproject.stanford.edu/software/invest, accessed on 10 May 2022) was used to calculate the water yield of the Yellow River Basin. It was calculated using the following equations [13,14]:

$$Y(x) = \left(1 - \frac{AET(x)}{P(x)}\right) \times P(x) \qquad (1)$$

where $AET(x)$ is the annual actual evapotranspiration for pixel $x$ and $P(x)$ is the annual precipitation on pixel $x$.

In fact, it is difficult to obtain the actual annual evapotranspiration at a large scale. In the InVEST model, potential evapotranspiration ($PET$) was used to calculate $AET$. $AET$ was easy to calculate based on $PET$ [15], which was estimated as the product of the reference

evapotranspiration and the crop coefficient for each grid square. The specific equation was listed as follows [16]:

$$\frac{AET(x)}{P(x)} = 1 + \frac{PET(x)}{P(x)} - [1 + (\frac{PET(x)}{P(x)})^w]^{1/w} \tag{2}$$

$$w = Z\frac{AWC(x)}{P(x)} + 1.25 \tag{3}$$

where $\omega$ is calculated based on the approach thorough the plant available water content (*AWC*), precipitation, and the empirical constant Z [17,18]. Z is an empirical constant, which reflects local hydrogeological properties, with a range of 1–30. *AWC* represents vegetation available water content, which can be calculated through soil texture and effective soil depth. For more details, see the InVEST 3.2.0 Users Guide and previous studies [17,19,20]. In the InVEST model, biophysical parameters were needed, including root restricting layer depth (mm), *AWC*, MAP, LULC, *PET*, and plant evapotranspiration coefficient (Kc). Relevant basic parameter used in the model of water yield in InVEST are listed in Table 2.

**Table 2.** Model parameters required to calculate water yield according to the InVEST model.

| Lucode | LULC_Desc | Kc | Root_Depth | LULC_Veg |
|:---:|:---:|:---:|:---:|:---:|
| 1 | Farmland | 0.65 | 2100 | 1 |
| 2 | Forest | 1 | 5300 | 1 |
| 3 | Grassland | 0.65 | 2400 | 1 |
| 4 | Water | 1 | 100 | 0 |
| 5 | Construction land | 0.3 | 100 | 0 |
| 6 | Unuse land | 0.5 | 100 | 0 |

Lucode: land use number; LULC_desc: land use types; LULC_veg: 1 means vegetation area, and 0 means no vegetation areas; Kc: plant evapotranspiration coefficient; Root_depth: the maximum root depth.

### 2.3.1. Water Demand

In this study, water demand was calculated by the water consumption of anthropogenic activities, including four parts: livestock water ($W_{liv}$), domestic water ($W_{dom}$), agricultural water ($W_{agr}$), and industrial water ($W_{ind}$) [21]. The specific equation was listed as follows:

$$Wu_x = W_{agr} + W_{liv} + W_{dom} + W_{ind} = AgeaAgr \times WaterAgr + Live \times Livewater + Popu \times Dom + GDP \times Ind \tag{4}$$

$W_{agr}$ is the consumption of agricultural water, which was calculated by multiplying the area of agricultural lands by the average irrigation water per hectare. $W_{liv}$ refers to livestock water use, which was calculated by multiplying the number of livestock by the annual water consumption per head. $W_{dom}$ refers to domestic water use, which was calculated by multiplying the population by the annual water consumption per resident. $W_{ind}$ refers to the consumption of in industrial water use, which was calculated by multiplying the gross industrial production (*GDP*) by the annual industrial water per 10,000 *GDP*.

### 2.3.2. The Imbalance between Water Supply and Demand and Spatial Flow Process

Ecosystem service flow had been considered as a comprehensive method to reflect actual water provision service [22]. We used the mismatches (WSI) between the supply and demand of water resources, which could reflect conflicts between water supply and demand, calculated by the ratio (S:D) of water supply (S) to water demand (D). In this study, the logarithm transformation was used to improve the comparison of WSI among different sub-watersheds. In this study, we calculated two kinds of WSI, static ($WSI_{static}$) and flow $\left(WSI_{flow}\right)$ conditions. The specific equation was listed as follows [23]:

$$WSI_{static,i} = log10\left(\frac{S_i}{D_i}\right) \tag{5}$$

where $i$ is the number of sub-watersheds; $S_i$ is the water supply on sub-watershed $i$; and $D_i$ is the water demand on sub-watershed $i$. When $WSI > 0$, it indicates that water supply is much more than water demand (water surplus); when $WSI < 0$, water supply is less than water demand (water deficit). The static WSI ignored the inflow of upstream water.

The WSI with in flow condition was calculated as follows:

$$Flow_i = S_i - D_i \times WC \tag{6}$$

$$WSI_{flow,i} = log10\left(\frac{WS_i - Flow}{WU_i}\right) \tag{7}$$

where $WC$ refers the regional water use and water consumption coefficient, collected from the local water resources bulletin. $WSI_{flow}$ considered the water service flows upstream.

## 3. Results

### 3.1. Spatial Patterns of Water Supply in the Yellow River Basin

In 2010, the water supply in the Yellow River Basin showed an obvious spatial pattern, and the variations depended on land use types (Figure 3). The total water supply was $3.03 \times 10^{11}$ m$^3$, with a range of $3.29 \times 10^8$ m$^3$ to $7.35 \times 10^{10}$ m$^3$ for different sub-watersheds. Different sub-watersheds had different water supply, with the highest supply in sub-watershed 28. Sub-watershed 12 provided the lowest water resources ($3.29 \times 10^8$ m$^3$). A high heterogeneity occurred in the whole watershed. Most sub-watersheds with higher water supply were distributed in the middle and upper reached reaches of the Yellow River Basin, such as sub-watershed 28, 29, 26, and 20. The sub-watersheds distributed in the lower reaches (sub-watershed 4, 6, 11, and 12) showed lower water supply.

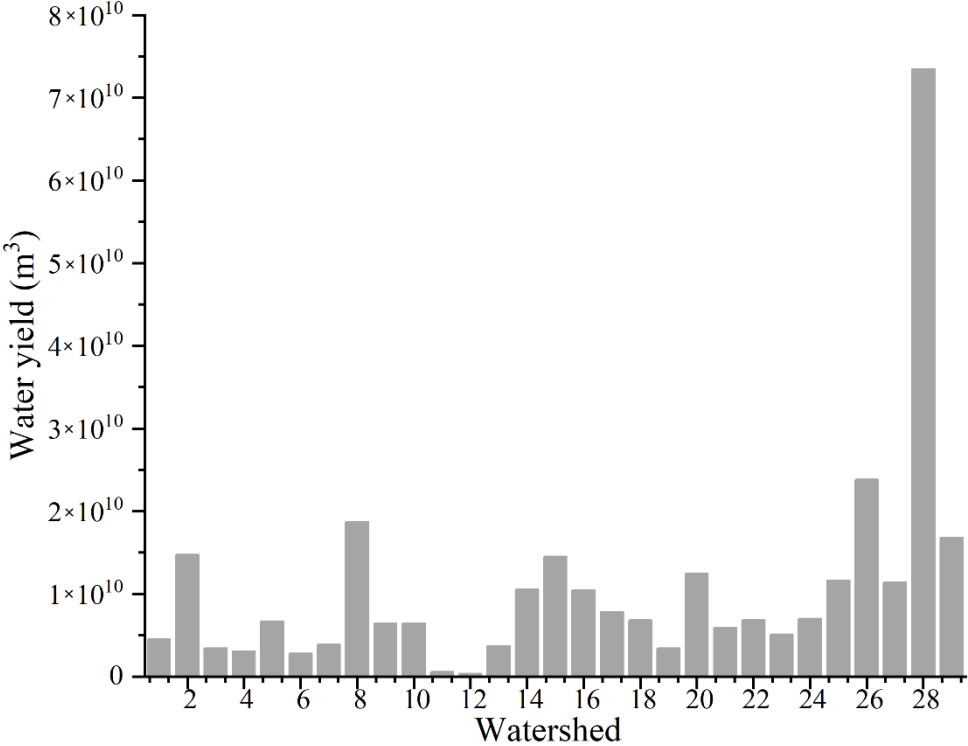

**Figure 3.** The water yield of different sub-watersheds in the Yellow River Basin.

Land use strongly affected the distribution of water supply in the Yellow River Basin. For example, forests and grasslands showed a low potential water supply, while constructed areas had relatively high potential in water supply. The land use spatial patterns suggested that constructed areas only accounted for 1%, which yielded relatively lower wa-

ter resources. Therefore, when accounting for the land use areas, the absolute water supply of grasslands was the most, followed by forests, agricultural lands, and unused lands.

### 3.2. Spatial Patterns of Water Demand in the Yellow River Basin

The Yellow River Basin supplied a total of $3.05 \times 10^{11}$ m$^3$ water resources, which was much more than the total water demand ($2.73 \times 10^{11}$ m$^3$). Although the water demand was less than water supply in the whole basin, the spatial distributions of water demand in sub-watersheds were not consistent with those in water supply (Figure 4). The higher water demands occurred in areas with high intensity industrial and urban areas, which were distributed in the upper reaches of the Yellow River Basin. The water demands were much higher in the middle and lower reaches than the upper reaches. The water demand exhibited a high spatial heterogeneity. For different sub-watersheds, watersheds 8, 6, 17, 18, 19, 20, 22, 23, and 26 showed relatively higher water demands, while watersheds 2, 3, 4, 5, 7, 9, and 28 had relatively lower water demands.

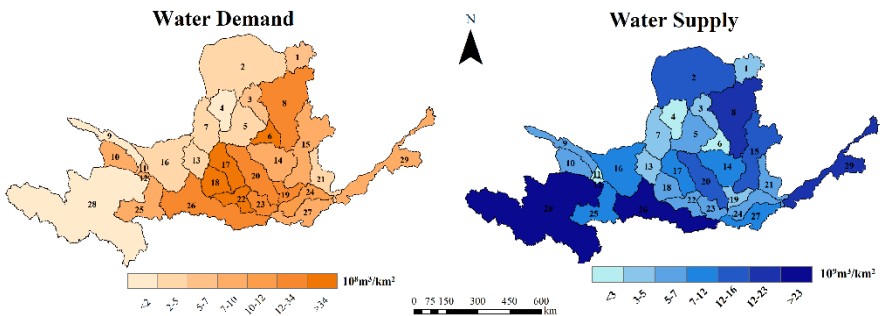

**Figure 4.** The spatial patterns of water supply and demand, and ecosystem service flow in the Yellow River Basin. The number in the figure is the number of the sub-watersheds.

When considering land use types, we found that Grasslands (404,071.8 km$^2$) and Unused land (19,845.9 km$^2$) showed a relatively lower water demand (Figure 5). Although the area of grassland was the most (49.7% of the total area), the water demand was the least ($3.21 \times 10^8$ m$^3$, 0.1% of the total water demand). Farmland contributed the most to the total water demand, accounting for 58.2% ($1.59 \times 10^{11}$ m$^3$), followed by construction area ($7.15 \times 10^{10}$ m$^3$), forest ($3.33 \times 10^{10}$ m$^3$), and water body ($9.11 \times 10^9$ m$^3$).

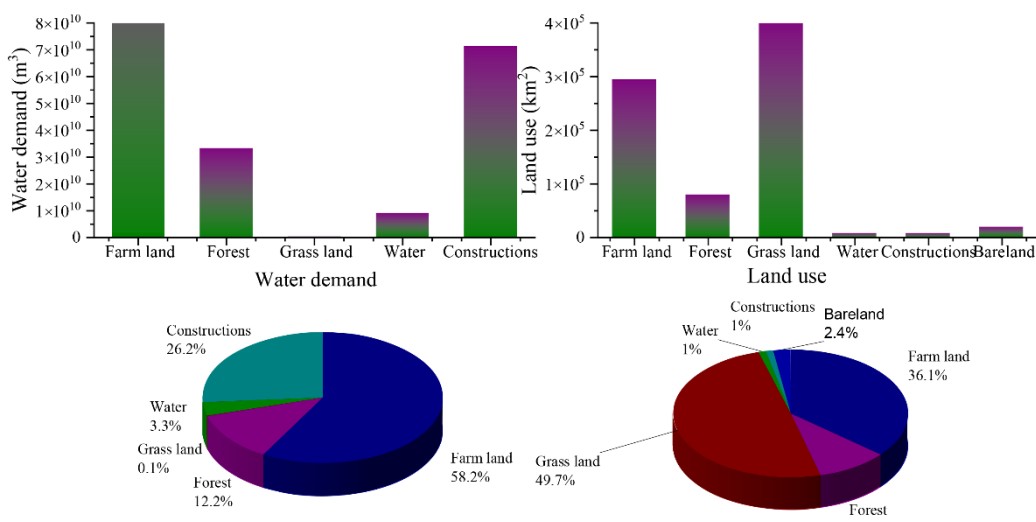

**Figure 5.** The proportion of different land use types and water demand in different land use type.

The Pearson correlation analysis revealed that water supply was mainly associated with farmland, grassland, and water areas, which significantly influenced water yields

(Figure 6). The total area of a city was significant correlated with water demand, which ignored the specific land types.

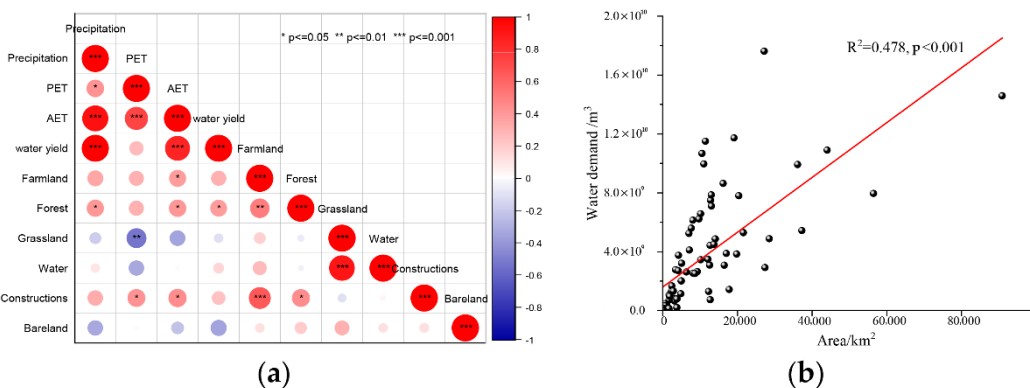

**Figure 6.** (**a**) The correlation of water yield between different land use areas, PET, AET, and precipitation; (**b**) the regression between the area of cities and water demand. Precip, precipitation; wyield, water yield.

### 3.3. Spatial Flow Process of Water Provision Service in the Yellow River Basin

We used WSI to clearly express the current situation of the conflicts between water supply and demand in the Yellow River Basin. At the static condition, WSI in most areas ranged from $-1$ to 0, and the values were different in the different sub-watersheds (Figure 7). Sub-watersheds 6, 11, 12, 17, 18, 19, 22, and 23 were the areas with the most shortage of water resources, and the scores of $WSI_{static}$ were $<-1$. Except for sub-watershed 9 and 28 with a higher $WSI_{static}$ ($>1$), $WSI_{static}$ in other sub-watersheds ranged from $-1$ to 1. The areas with relatively high $WSI_{static}$ were mainly distributed in the lower and upper reaches of the Yellow River Basin in sub-watersheds 28, 29, 2, and 9. When considering water flow from upstream to downstream, there was a slight difference in WSI in comparison with the static condition. For example, sub-watershed 29 changed from water surplus ($WSI_{static} > 0$) to water deficit ($WSI_{flow} < 0$).

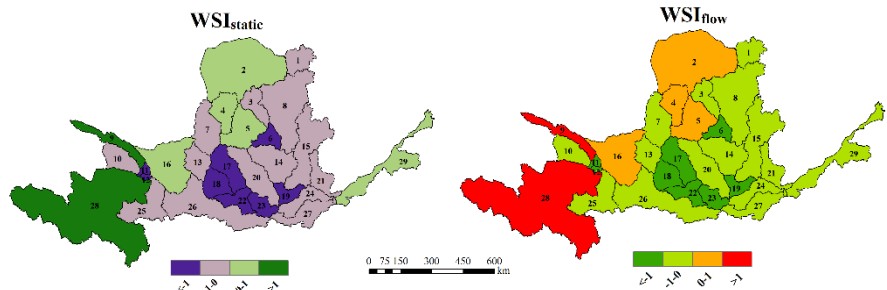

**Figure 7.** The ecosystem service flow of water resources ($WSI_{static}$ and $WSI_{flow}$) in the Yellow River Basin. The number in the figures is the number of the sub-watersheds.

## 4. Discussion

This study used the framework proposed by a previous study to assess the spatial patterns of water supply and demand, and the effects of land use on water supply–demand balance. The results showed obvious changes in water supply and demand among different sub-watersheds, and these changes were strongly dependent on land use types. We highlighted the roles of land use on spatial patterns of water supply and demand. We also used WSI to determine the flow process of water resources using the ratio of water supply to water demand, and the results suggested the water shortage areas were clustered in the lower reaches of the Yellow River Basin. When combined with land use changes, we found that urban construction, agricultural irrigation, population, and domestic life were

the main drivers in regulating water resources and the causes of conflicts between water supply and demand balance.

### 4.1. The Effects of Land Use on Water Supply and Demand Balance

A large body of studies has shown that land use is the most important factor in regulating global environmental change, and mitigating global warming, extreme rainfall and dry events, and water shortage [24–27]. In this study, we focused on the balance between water supply and demand in Yellow River Basin. We found that land use was the main driver in regulating water supply and demand. Differently, the contribution of specific land use to water supply was not consistent with water demand. For example, grassland contributed more to the total water supply of the Yellow River Basin, while urban area and agricultural lands contributed more to the total water demand. A previous study showed that grassland had the highest water supply because of the largest areas in the Yanhe watershed [10]. In this study, the area of grassland was $405,296 \times 10^4$ km$^2$, accounting for 49.7% of the total area. Therefore, grassland contributed the most to the water yield. Although the largest area of grassland, the water demand per ha was relatively low, suggesting a water surplus. Previous studies reported that most water supply occurred in urban areas [28], which was consistent in this study. The expansion of urban area would alter the local climate process and the magnitude of precipitation [29].

The water demand mainly includes industrial water, agricultural water, domestic water, and livestock water [30]. These activities mainly happen in urban and agricultural areas, as well as some grasslands and forests. This was the main cause for the mismatches between water supply and demand. Urban areas consume the most water, followed by agricultural lands, while the total supply in urban and agricultural areas were relatively low because of the lower areas. In the Yellow River Basin, most areas were grasslands and forests, which consumed relatively low water resources. The differences of water demand in different sub-watersheds were mainly caused by land use changes and human social activities. For example, sub-watersheds 2, 4, 5, 9, and 28 were mainly located in natural vegetation areas (grassland and forest) with less population, which need less water resources. The water supply would therefore satisfy the demand.

### 4.2. The Spatial Flow Process of Water Resources in the Yellow Water Basin

The mismatches and the ratio of the supply to demand have been considered as important indicators to evaluate the balance between water supply and demand [11,21,31]. A positive SD ratio means a water service surplus, a null SD ratio suggests a supply–demand balance, and a negative SD ratio reveals a water deficit. The mismatch between the supply and demand of water resources occurred in most areas as reported in previous studies [32–34]. The mismatch reflects the results caused by the interaction of climate, terrain, socio-economic characteristics, land uses, and human activities. For example, the rapid urban expansion caused the shortage of water resources and imbalance of water supply–demand [33,35,36]. Scientific assessment of the water provision service and the supply–demand balance could help determine the development of urban construction and population planning, as well as provide constructive suggestions for government officials and urban landscape policy makers [36].

The spatial heterogeneity in the whole basin caused mismatches between water supply and demand. The water supply exhibited a decreasing trend from west to east, and western areas had high water supply, while areas with higher water demand were in the upper and lower reaches of the whole basin. This mismatch was mainly associated with land use. The flow process decoupling between water supply–demand balance and human activities enhance the understanding of water flow process in different areas. Water service flow reflects the processes between water supply and demand, including complex processes [37]. This study focused on the water flow process in different sub-watersheds of the Yellow River Basin and highlighted different spatial patterns in different sub-watersheds in comparison with the whole watershed. These changes of the water flow process indicate

the importance of sub-watersheds in assessing waster provision service. The imbalance between water supply and demand based on a sub-watershed would provide more precise details in regulating water resources. For example, a previous study focused on urban areas showed water yield exhibited an excess of supply in Hangzhou city [38]. In addition, the water flow process of different sub-watersheds could provide specific strategies to manage the limited water resources. Therefore, this study provides more useful details in assessing water provision service and exploring the underlying mechanisms, which precisely help policy makers to manage water resources and be references for other ecosystem services evaluations.

## 5. Conclusions

This study used the InVEST model to evaluate spatial patterns of water supply and demand in the Yellow River Basin and then calculated water spatial flow process depending on land use. The S:D ratio (WSI) was used to assess the spatial patterns of the imbalance between water supply and demand. Our results showed inconsistent changes for water supply and demand resulting in spatial mismatches and an imbalance between water supply side and demand side. The higher water supply potential was distributed in the middle and upper reaches. These changes in water supply are mainly caused by LULC changes caused by human activities; for example, unreasonable land use conversion, industrialization, and urbanization. Our results also suggest that water demand and water spatial flow process need to be considered in assessing water provision services. A water flow process would help land policy makers and local government to optimize land managements and reduce the deficit of water resources.

**Author Contributions:** Conceptualization, Y.L. and S.A.; methodology, Y.L.; data analysis, Z.W. and Y.L.; writing—original draft preparation, Y.L., S.A. and Y.Y.; writing—review and editing, Y.L. and S.A.; project administration, S.A. All authors have read and agreed to the published version of the manuscript.

**Funding:** This research was funded by the National Natural Science Foundation of China, grant number 42077072, the Open Program of State Key Laboratory of Soil Erosion and Dryland Farming on The Loess Plateau, Institute of Soil And Water Conservation, CAS&MWR, grant number A314021402-1812, and the International Partnership Program of Chinese Academy Of Sciences, grant number 161461KYSB20170013.

**Institutional Review Board Statement:** Not applicable.

**Informed Consent Statement:** Not applicable.

**Data Availability Statement:** Not applicable.

**Conflicts of Interest:** The authors declare no conflict of interest.

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
