# Peer review of "Quantifying Water Provision Service Supply, Demand, and Spatial Flow in the Yellow River Basin"

_sustainability, doi:10.3390/su141610093_

Round 1

Reviewer 1 Report

Strong research contribution element is lacking..other than water balance computation of a region using an existing procedure, some research contribution is expected and stated clearly

This study used the framework proposed by previous study to assess- not clear revise

Add equation numbers

Line 273 etc references  use numbering .. check at other places
Subscripts and superscripts – please check throughout

L327 : of water provision ecosystem service . not clear

Basically demand and supply are considering and water balance is quantified  it seems ..spatial flow is confusing

WSI is to be mentioned  .. Line 120 it is not clear..expand

Figure 6 (a) and (b) should be specified

Here relationship of S/D with LU only is considered .. what are the other parameters  affecting that to be elaborated .. in this context L267 is a misfit

What about the role of human interventions  in the form of  flow regulation/structures ?

Author Response

Thanks for your constructive suggestions which help improve the quality of this Manuscript.

1.Strong research contribution element is lacking. Other than water balance computation of a region using an existing procedure, some research contribution is expected and stated clearly

Response: This study provides a scientific basis for watershed policymakers to optimize land management and reduce water deficits and mismatches.

2.This study used the framework proposed by previous study to assess- not clear revise

Response: The framework was used based on a previous study in Yanhe watershed, which was belonged to the Yelllow River Basin. 

3.Add equation numbers

Response: We have added the number of the equation.

4.Line 273 etc references use numbering . check at other places

Response: We have revised this error and used the number style and checked throughout the manuscript.

5.Subscripts and superscripts – please check throughout

Response: We have revised the subscripts and superscripts throughout the manuscript.

6.L327 : of water provision ecosystem service . not clear

Response: We used the imbalance between water supply and demand.

7.Basically demand and supply are considering and water balance is quantified it seems. spatial flow is confusing

Response: The essence of the research on water supply service flow is to quantitatively evaluate the supply and demand of water provision service and to establish the space-time relationship between supply and demand.

8.WSI is to be mentioned. Line 120 it is not clear. expand

Response: We used the mismatches (WSI) between the supply and demand of water resources, which could reflect the conflicts between water supply and demand, calculating by the ratio of water supply (S) to water demand (D) (S:D).

9.Figure 6 (a) and (b) should be specified

Response: The correlation of water yield between different land use areas, PET, AET and precipitation (a); the regression between the area of watersheds and water demand (b).

10.Here relationship of S/D with LU only is considered. what are the other parameters affecting that to be elaborated .. in this context L267 is a misfit

Response: Land use change mainly caused by intensive anthropogenic activities, which was a comprehensive factor associated with population and economic factors.

11.What about the role of human interventions in the form of flow regulation/structures?

Response: Land use changes mainly caused by intensive anthropogenic activities have the strongest effects on both the supply side and the demand side.

Reviewer 2 Report

Manuscript. Number.: Sustainability-1803568

Title: Quantifying water provision service supply, demand, and spatial flow in the Yellow River Basin

 Abstract

1.      It is necessary to rewrite the abstract section. You should include more details than the results of your research. Some quantitative results should be included.

Introduction

2.      What is the main innovative aspect of the research study? How is it different and important from similar studies?

Study area

3.      The map of the study area should be drawn more clearly and in detail at the scale of the basin. Line 95.

Data collection

4.      When writing some words briefly for the first time, it is better to write them briefly in parentheses after writing a detailed description, and then after writing them in abbreviated form. For example, Line 104 DEM.

Data analysis

5.      Write numbers behind the equation.

            Line 125, 133, 134,156-159, 177, 182-184

6.      Draw and change Figure 2 in more detail

Results

7.      Improve Figures 3 and 4.

8.      Write down the size of the area in terms of KM2. Line 224-226.

Conclusion

9.      Rewrite the conclusion section?. The logic is out of order. Some quantitative results should be included.

10.  Write down the shortcomings of the study.

Author Response

Thanks for your constructive suggestions which help improve the quality of this Manuscript.

Abstract

1.It is necessary to rewrite the abstract section. You should include more details than the results of your research. Some quantitative results should be included.

Response: We have revised the abstract and added some quantitative results.

Introduction

2.What is the main innovative aspect of the research study? How is it different and important from similar studies?

Response: This study considered water demand and flow process of water resources between supply and demand, which was different from most studies. Most studies mainly assess the relationship between water supply and demand under a non-flowing condition that does not consider the influences of upstream water. Therefore, there is a lack of research on the spatial flow path of water provision service and information relevant to decision-making.

Study area

3.The map of the study area should be drawn more clearly and in detail at the scale of the basin. Line 95.

Response: We have added the details of the Yellow River Basin.

Data collection

4.When writing some words briefly for the first time, it is better to write them briefly in parentheses after writing a detailed description, and then after writing them in abbreviated form. For example, Line 104 DEM.

Response: Thanks for your suggestions. We have added the detailed description of these abbreviation.

Data analysis

5.Write numbers behind the equation. Line 125, 133, 134,156-159, 177, 182-184

Response: We have added the numbers of these equations.

6.Draw and change Figure 2 in more detail

Response: We have added the details of Fig. 2.

Results

7.Improve Figures 3 and 4.

We have revised Fig. 3 and Fig. 4.

  1. Write down the size of the area in terms of KM2. Line 224-226.

Response: We have revised this part.

Conclusion

9.Rewrite the conclusion section? The logic is out of order. Some quantitative results should be included.

Response: Thanks for your suggestions. We have revised this section and added some quantitative results.

10.Write down the shortcomings of the study.

Response: We have added the shortcomings of this study. This study didn’t include other factors such as population, economic factor, and the specific human activity. In addition, the main source of uncertainty in quantifying water demand with statistical data is the limitation of data acquisition, which exacerbate the quantitative deviations in water demand.

Round 2

Reviewer 1 Report

Nil

Author Response

Thanks for your constructive suggestions which help improve the quality of this Manuscript.

Reviewer 2 Report

Manuscript. Number.: Sustainability-1803568

Title: Quantifying water provision service supply, demand, and spatial flow in the Yellow River Basin

No improvement has been made on the following 5 items.

Please do it again!

Minor revisions are needed for this manuscript to be published in the Sustainability.

Introduction

1.      What is the main innovative aspect of the research study? How is it different and important from similar studies?

Study area

2.      The map of the study area should be drawn more clearly and in detail at the scale of the basin. Line 95.

Data analysis

3.      Draw and change Figure 2 in more detail

Results

4.      Improve Figures 3 and 4.

5.      Write down the size of the area in terms of km2. Line 224-226.

Author Response

Minor revisions are needed for this manuscript to be published in the Sustainability.

Thanks for your constructive suggestions which help improve the quality of this Manuscript. All the revised parts were in blue color.

Introduction

  1. What is the main innovative aspect of the research study? How is it different and important from similar studies?

Response: This study considered water demand and flow process of water resources between supply and demand, which was different from most studies. Most studies mainly assess the relationship between water supply and demand under a non-flowing condition that does not consider the influences of upstream water. Therefore, there is a lack of research on the spatial flow path of water provision service and information relevant to decision-making. See line 70-73 and 75-79.

Study area

  1. The map of the study area should be drawn more clearly and in detail at the scale of the basin. Line 95.

Response: Figure 1. The location and the sub-watersheds of the Yellow River Basin. The number is the sub-watershed in the whole river. We have added the sub-watersheds in Fig.1.

Data analysis

  1. Draw and change Figure 2 in more detail

Response: Figure 2 showed the basic characteristics of the Yellow River Basin, which were used to calculate the water yield in Invest Model.

Results

  1. Improve Figures 3 and 4.

Response: We have revised figure 3 and 4. Fig. 3 showed the distributions of water yield in different watersheds according to a histogram. The Y-axis had been revised using scientific style. Fig.4 has been divided into two figures and the color was easier to grasp the details about water demand and water supply in different watershed.

  1. Write down the size of the area in terms of km2. Line 224-226.

Response: We have added the details of each area about the water demand in Fig.4.
